# Papillomaviruses in Domestic Cats

**DOI:** 10.3390/v13081664

**Published:** 2021-08-22

**Authors:** John S. Munday, Neroli A. Thomson

**Affiliations:** Pathobiology, School of Veterinary Science, Massey University, Palmerston North 4410, New Zealand; n.thomson1@massey.ac.nz

**Keywords:** cats, felid, papillomavirus, review, cancer, viral oncogenesis, skin

## Abstract

Papillomaviruses (PVs) are well established to cause hyperplastic papillomas (warts) in humans and animals. In addition, due to their ability to alter cell regulation, PVs are also recognized to cause approximately 5% of human cancers and these viruses have been associated with neoplasia in a number of animal species. In contrast to other domestic species, cats have traditionally been thought to less frequently develop disease due to PV infection. However, in the last 15 years, the number of viruses and the different lesions associated with PVs in cats have greatly expanded. In this review, the PV life cycle and the subsequent immune response is briefly discussed along with methods used to investigate a PV etiology of a lesion. The seven PV types that are currently known to infect cats are reviewed. The lesions that have been associated with PV infections in cats are then discussed and the review finishes with a brief discussion on the use of vaccines to prevent PV-induced disease in domestic cats.

## 1. Introduction

Papillomaviruses (PVs) are double-stranded circular DNA viruses. Their genome contains five or six early (E) genes and two late (L) genes. With rare notable exceptions, PVs are species specific and PVs often show tropism for certain types of epithelium and even specific locations on the body [1]. PVs are classified using the highly conserved L1 gene. If two PVs have 60–90% similarity in the L1 open reading frame (ORF), then they are considered to be different types while less than 60% similarity suggests that the PVs are likely to be within different genera [2]. Members of a genus often infect closely related host species and often result in similar lesions within that host [2]. Papillomaviruses have been found to infect almost all species that have been studied including mammals, birds and reptiles [3]. Most species are infected by multiple PV types, often within multiple different genera [2]. 

The PV life cycle is coordinated with the normal division and differentiation of cells within mucocutaneous stratified epithelium [4]. Microtrauma initially allows the PV to gain access to the basal cells. Expression of the PV E1 and E2 genes results in the virus creating a small number of copies of itself which then infect surrounding basal cells [4]. The infection of basal cells allows persistence of PV infection, but viral replication is only possible when a basal cell terminally differentiates and moves into the suprabasilar layer of the epithelium [4]. Here, due to the expression of the E6 and E7 proteins, the PV interferes with cell regulation by preventing the cells from terminally differentiating, ensuring the nucleus is retained, and forcing the epithelial cells to divide and make copies of the PV [4]. As the infected cells near the surface of the epithelium, the L1 and L2 proteins are expressed, allowing the virion to be assembled. Cells slough from the surface of the epithelium and normal epithelial cell degeneration releases the viral particles into the environment [4]. 

Whether or not infection by a PV will result in a visible lesion is dependent on the amount of epithelial replication (and thus viral replication) that the PV is able to stimulate. For the majority of PV infections, the virus replicates slowly, resulting in a mild increase in epithelial cell replication which is not detectible clinically [5]. Such asymptomatic infections appear to be ubiquitous in humans and are probably also extremely common in the domestic species [6]. In contrast, a minority of PV types rapidly replicate and stimulate marked epithelial replication. The resultant thickening and folding of the epithelium is visible as a hyperplastic viral papilloma (wart) [1,7]. Warts develop due to rapid viral replication and massive numbers of viral particles are produced in the affected areas of epithelium. 

As PV infection does not cause cell necrosis and the majority of the effects of the virus are in the more superficial layers of the epithelium, PVs tend to stimulate a mild inflammatory response. This is especially true for the PVs that replicate slowly and are asymptomatic [8]. When a response is made by the body, the body detects and attacks infected cells by developing a cell-mediated immune response [9]. This immune response is able to limit PV replication and, due to the loss of PV proteins influencing cell growth, any hyperplastic lesion induced by the infection will resolve. The time of onset of the cell-mediated immune response is variable [9]. This variability explains why, although most oral papillomas in dogs will spontaneous resolve in 3 months, others will persist for up to a year [10]. However, while the immune response results in lesion resolution, PVs are able to persist in the basal cells and probably continue to replicate at a low rate. The role of the immune system in controlling PV replication is demonstrated by the lack of any visible lesions due to the ubiquitous infection of the skin by the human *betapapillomaviruses* in immunocompetent people [6]. However, these same viruses are able to cause multiple hyperplastic plaques on the skin when people are immunosuppressed [11]. Likewise, chronic immunosuppressive therapy in dogs has been reported to predispose to PV-induced pigmented plaques of the skin [12]. Infection by a PV also results in the production of serum antibodies. These do not appear to influence lesion resolution, but protect against subsequent infection by that PV type [13]. 

In addition to the development of hyperplastic papillomas, the ability of PVs to alter cell regulation means that they can also influence the development of cancer [4]. In people, PVs are the most common viral cause of cancer with the high-risk *alphapapillomaviruses* causing approximately 5% of all human cancers including most cervical squamous cell carcinomas (SCCs) as well as a significant proportion of oral SCCs [14]. Likewise in the domestic species, PVs have been associated with neoplasia in horses, dogs, cattle, pigs, and sheep [15,16,17,18,19]. However, it is important to recognize that the vast majority of PV infections in humans and animals do not result in the development of neoplasia. It appears that additional factors such as the speed of the immune response or the presence of other promotors of neoplasia are critical in determining whether or not a PV infection will result in cancer [8,20]. 

As in other species, the majority of cats are infected by PVs [21]. However, PV-induced disease appears to be rare in cats compared to the other domestic species [22]. This review briefly discusses the methods used to investigate a potential PV etiology within a lesion. The seven fully sequenced PV types that are currently recognized to infect cats are then reviewed. The diseases associated with PVs in cats are described and the review ends with the potential to use vaccines to prevent PV-induced diseases in cats.

## 2. Methods Used to Investigate a Potential PV Etiology 

### 2.1. Histology

Histology can be used to diagnose lesions that have been previously established to be caused by PV infection. Virally-induced oral papillomas, Bowenoid in situ carcinomas (BISCs) and feline sarcoids all have characteristic histological features that will allow a diagnosis of a PV-induced lesion. In addition, oral papillomas and BISCs may contain histologically detectible changes within the epithelial cells caused by PV replication. As each PV type results in slightly different changes in the epithelial cells, the changes are reported within the descriptions of each Felis catus papillomavirus (FcaPV) type (see below). Intranuclear inclusions may also be present but can be difficult to differentiate from nucleoli. Viral replication appears to decrease as lesions progress from hyperplasia to neoplasia and histologically visible cell changes due to PV replication are not usually present in advanced BISCs or SCCs. 

### 2.2. Immunohistochemistry

There are two ways that immunohistochemistry (IHC) can be used to investigate a PV etiology of a lesion. Firstly, IHC can be used to detect PV L1 antigen within a lesion. Localization of PV antigen within the proliferating cells provides convincing evidence that a hyperplastic lesion was caused by PV infection. However, as L1 antigen is only produced in the late stages of viral replication, lesions that do not contain viral replication will not contain L1 antigen [1]. Viral replication typically causes histologically identifiable PV-induced cell changes. Therefore, in the authors’ experience, L1 antigen is rarely detectible in lesions that do not also contain PV-induced cell changes. In addition, there are currently no antibodies available against feline PV types. As a lack of cross-reactivity between the antibodies and the feline PV L1 protein is possible, it is hard to interpret negative results. 

Viral replication is rare within PV-induced cancers. Therefore, cancers typically do not contain immunostaining using anti-PV L1 antibodies and these antibodies are rarely used in human pathology. Instead, immunohistochemical detection of increased p16^CDKN2A^ protein (p16) is used as a proxy marker of a PV etiology of a SCC [23]. Neoplasms that are caused by PVs contain increased p16 due to the consistent PV-induced degradation of the retinoblastoma protein (pRb). As p16 prevents cell division by inhibiting pRb, loss of pRb results in a marked accumulation of p16 within the cell [24]. Using IHC to detect the accumulated p16 allows the changes caused by the PV proteins to be visualized rather than the PV itself. This means that evidence of PV infection will remain even in cancers where PV replication is no longer present. Immunohistochemistry for p16 can be used with confidence in cats as the G175-405 clone (although not necessarily other clones) of the p16 antibody reliably cross-reacts with the feline p16 protein [25]. A disadvantage of p16 immunostaining is that the sensitivity and specificity of this for PV infection is unknown in cats. While studies show a strong association between the presence of PV DNA and p16 immunostaining [25,26] a spontaneous mutation in pRb would also increase p16 independently of any role of the PV in cancer development. Likewise, a spontaneous mutation in p16 would reduce the presence of p16 in a lesion, even if PV infection had caused degradation of pRb. 

### 2.3. Molecular Techniques

The presence of PV DNA within a lesion can also be investigated using molecular techniques. This can be done using PCR to amplify sections of PV DNA or RNA from formalin-fixed or unfixed tissue samples. Specifically-designed PCR primers can be used if the causative PV type is known or, if the PV type is unknown, consensus PCR primers which have been designed to amplify a wide range of PV types can be used [6,27]. PCR has the advantage of being able to detect very few viral copies and detection is not dependent on the presence of viral replication. Quantitative PCR can be used to determine the amount of viral DNA within a lesion. However, a disadvantage of using PCR to detect viral DNA is that the location of the viral DNA or RNA within the lesion cannot be determined. As it is known that many animals are asymptomatically infected by PVs, amplifying viral DNA or RNA from a lesion is not definitive proof that the lesion was caused by the PV. To overcome this limitation, more recent studies have used in situ hybridization (ISH) to localize the PV DNA or RNA within the lesion [28,29,30]. Demonstrating that PV DNA and RNA is present within the basal and suprabasilar layers of a lesion provides good evidence supporting a role of the virus in the development of that lesion.

## 3. Papillomaviruses That Infect Domestic Cats

### 3.1. Felis catus Papillomavirus Type 1

The first full PV sequence from a domestic cat was published in 2002 [31] (Figure 1). This PV type was originally designated Felis domesticus PV (FdPV) 1; however, as the correct taxonomic name for domestic cats is *Felis catus*, PVs identified from this species have been renamed Felis catus PVs (FcaPVs). Felis catus PV type 1 is the only *lambdapapillomavirus* known to infect domestic cats. However, this virus is closely related to the *lambdapapillomaviruses* that infect exotic felids [3]. As there are only rare reports of FcaPV1, the age at which cats are infected and the proportion of cats that are infected are both unknown. Although FcaPV1 was initially detected in a skin lesion, this virus has subsequently only been detected within the oral cavity [32,33]. 

Evidence from the small number of FcaPV1-associated lesions that have been described suggests infection of a cell by FcaPV1 results in expansion of the cytoplasm and the presence of characteristic eosinophilic block-shaped cytoplasmic bodies [32]. The nuclei of affected cells contains chromatin that is compressed peripherally with an area of central nuclear clearing that often contain a prominent nucleoli (Figure 2).

### 3.2. Felis catus Papillomavirus Type 2

Of the PVs that infect domestic cats, FcaPV2 appears to be the most common cause of disease and much of the research into PVs in cats has focused on this virus. The full sequence of FcaPV2 was reported in 2009 and this PV was classified as the only member of the *dyothetapapillomavirus* genus [34]. Cats are initially infected by FcaPV2 within the first few days of life, probably from the queen during birth or from close contact during suckling and grooming [21,35]. Infection by FcaPV2 is very common. Viral DNA can be detected in skin swabs from almost all clinically normal cats [36,37,38] and serum antibodies against FcaPV2 are detectable in a quarter of cats [38]. In addition, FcaPV2 DNA and gene expression were detected in blood samples from clinically healthy cats suggesting viral replication in non-epithelial cells and the potential for viral transmission through the blood or placenta [39]. 

The overwhelming majority of FcaPV2 infections are asymptomatic. However, in a small proportion of cats, FcaPV2 is able to induce sufficient epithelial hyperplasia to cause visible skin lesions. Furthermore, these lesions often contain evidence of loss of normal cell regulation and these areas of virally-induced hyperplasia can progress to invasive neoplasia. Currently it is poorly understood why FcaPV2 infections in some cats result in clinical disease. However, considering the ubiquitous infection of cats by this virus, host factors appear likely to be most important in determining the development of disease. Interestingly, the viral load on an individual cat appears to remain consistent over time, suggesting the viral load on a cat is determined by the genetics of that individual [37]. Additionally, some breeds of cats appear particularly susceptible to FcaPV2-induced disease [40,41]. This suggests that the development of disease due to FcaPV2 is most likely due to a genetically-determined immune response to infection by this PV. 

Felis catus PV type 2 has been shown to have the potential to influence normal cell regulation through multiple pathways. Studies on human cancers show that PVs influence cancer development by degrading the retinoblastoma protein (pRb) [4]. As pRb is a key gatekeeper in the cell replication pathway, loss of this protein promotes cell division. In addition, loss of pRb results in high levels of p16 within a cell which can be detected by IHC [23,24]. Similarly in cats, pRb was typically undetectable, but intense p16 immunostaining was present in lesions that contained FcaPV2 DNA [42,43,44]. Subsequent studies confirmed that the FcaPV2 E7 protein can bind to pRb [45] and evidence currently suggests that FcaPV2 is able to promote cell division by E7-protein mediated degradation of pRb within the cells. The human ‘high risk’ PVs also promote cancer by degrading the p53 protein [4]. No association between the presence of FcaPV2 DNA within a lesion and the presence of absence of p53 protein was observed in initial IHC studies [46]. However, more recent molecular techniques showed that the FcaPV2 E6 protein can promote degradation of p53 within the cell suggesting this PV may also influence neoplasia by interfering with normal p53 function [45,47]. In addition, FcaPV2 may also promote cell replication by increasing the expression of mitogen-activated protein kinases and inhibiting cell apoptosis by increasing the expression of protein kinase B [48]. 

While there is good evidence that FcaPV2 causes hyperplastic lesions of the skin of cats, it is currently unknown how much the PV infection influences the progression to neoplasia. In humans, additional spontaneous mutations are required before a PV-induced lesion progresses to cancer [4]. However, as the PV increases cell replication and inhibits apoptosis, the likelihood of spontaneous mutations developing is greatly increased [35]. It may be similar in cats with additional factors required before a PV-induced hyperplastic lesion progresses to neoplasia. Exposure to UV light damages cell DNA and this could be a cofactor for some skin cancers in cats. In normal skin, most UV-induced DNA damage is repaired prior to cell replication. However, it is possible that when UV-induced DNA damage occurs within a PV-induced lesion, the presence of the PV induces the cell to replicate and inhibits apoptosis, predisposing to cancer development [45,49]. 

Of all the FcaPV types, the histological cell changes associated with FcaPV2 have been best characterized. Cell changes due to this PV include expansion of the cytoplasm with either non-staining material or with wispy, granular or amorphous blue-grey material [50]. Unlike FcaPV types 1 and 3, FcaPV2 does not result in the development of any intracytoplasmic bodies. Moderately enlarged cells with dark shrunken nuclei that are surrounded by a clear halo (koilocytes) are often visible in the deeper layers of the affected epidermis. 

### 3.3. Felis catus Papillomavirus Type 3

The complete sequence of this PV type was first reported in 2013 [51]. The L1 ORF sequence of FcaPV3 has less than 60% similarity to both FcaPV1 and FcaPV2, but was found to be most similar to canine PVs within the *Taupapillomavirus* genus. Due to genetic and behavioral similarities between FcaPV3 and the canine PVs, this PV was classified as the first member of the species 3 *Taupapillomaviruses* [52]. This PV has subsequently been detected within hyperplastic and neoplastic skin and oral lesions in cats, suggesting the PV can alter cell regulation [53,54,55]. Additionally, lesions that contained FcaPV3 DNA had marked p16 immunostaining suggesting that the virus can degrade pRb [53]. It is currently unknown how many cats are asymptomatically infected by FcaPV3 and when cats are first infected by this PV type. 

Infection by FcaPV3 has been reported to induce some characteristic cell changes [53]. These include cells with large quantities of non-staining cytoplasm and the presence of elongated amphophilic to basophilic perinuclear intracytoplasmic bodies. 

### 3.4. Felis catus Papillomavirus Type 4

The complete sequence of this PV was reported in 2014 after being detected in a sample from the mouth of a cat [56]. The L1 ORF of this virus is most similar to FcaPV3 and this virus is also classified as a species 3 *Taupapillomavirus* [52]. This virus has also been rarely detected in cutaneous lesions in cats [29,54,57]. As FcaPV4 was the only PV type detected in some lesions, this suggests that this PV is able to cause disease in cats. Additionally, the presence of FcaPV4 DNA has been associated with intense p16 immunostaining suggesting that this PV type can increase cell replication by interfering with normal pRb function [25]. The proportion of cats infected by this PV type and the age at infection are unknown. 

As there are only a few lesions that have been described due to FcaPV4 in cats, it is difficult to be definitive regarding the cell changes that develop as a result of infection by this virus. However, in the limited number of lesions examined by the authors, a high proportion of the cells within the area of hyperplasia contained large quantities of finely granular blue-grey cytoplasm and peripherally-displaced cell nuclei [56]. Cytoplasmic bodies have not been visible in the cells of lesions associated with FcaPV4. 

### 3.5. Felis catus Papillomavirus Type 5

This papillomavirus type was reported in 2017 after being detected in a skin lesion [58]. Although this PV has not yet been officially classified within a genus, the sequence of this PV is most similar to FcaPV3 and FcaPV4, suggesting that this PV will also be classified as a species 3 *Taupapillomavirus*. Since the initial report of this PV type, there have been rare additional reports in which FcaPV5 has been detected in skin lesions in cats [29,59]. Lesions associated with FcaPV5 contained p16 immunostaining suggesting that this virus can influence cell growth by interfering with normal pRb function [60]. 

This virus has been associated with a limited number of lesions. However, current evidence suggests that this PV type can infect cells deep within the epidermis including the adnexal structures [60]. Therefore, infection with FcaPV5 should be suspected when apocrine or sebaceous gland cells contain cytoplasm that is expanded by blue-grey cytoplasm. Intracytoplasmic bodies have not been observed in association with FcaPV5, but cells with large clear vacuoles may be indicative of infection with this PV type. 

### 3.6. Felis catus Papillomavirus Type 6

The most recent PV type to be fully sequenced from domestic cats was also detected in a skin lesion and was reported in 2020 [61]. The L1 ORF sequence of this PV type is most similar to FcaPV3 suggesting that this PV will also be classified as a species 3 *Taupapillomavirus*. Felis catus papillomavirus type 6 has not been subsequently reported, suggesting that this PV type may be a rare cause of skin disease in domestic cats. 

The lesion that contained FcaPV6 did not contain any PV-induced cell changes so histological features of infection by this type are unknown [61]. 

### 3.7. Bovine Papillomavirus Type 14

While this PV type is able to infect cats, cattle are the definitive hosts of BPV14. In cattle, this virus has been detected within papillomas (warts), bladder cancers, and in samples of normal skin [62,63,64]. In contrast, BPV14 has only been detected in samples of sarcoids (as described later, a type of mesenchymal neoplasia) in cats. This PV does not asymptomatically infect cats and cats appear to be dead-end hosts for this virus [65]. In addition to domestic cats, BPV14 has also been detected in sarcoids from African lions and cougars [66,67]. 

Bovine PV type 14 is classified as a *deltapapilloavirus* and is most closely related to BPV1, 2 and 13, the causes of bovine papillomas and equine sarcoids [68,69]. The bovine *deltapapillomaviruses* are highly unusual in their ability to cause proliferation of both mesenchymal and epithelial cells in cattle as well as mesenchymal cell proliferation in non-bovine species [20]. Interestingly, while the BPV types that cause equine sarcoids are closely related to BPV14, BPV14 does not appear to cause equine sarcoids and neither BPV1 nor 2 have been reported in a feline sarcoid. This suggests that although these BPV types are able to cause cross-species infection, they are only able to infect a limited number of species [20]. 

## 4. Lesions Associated with Papillomaviruses in Domestic Cats

### 4.1. Cutaneous Papillomas (Warts)

Warts develop due to marked thickening and folding of the epidermis due to PV infection. As the marked epidermal hyperplasia allows massive viral replication, histological evidence of PV replication is prominent. Warts are hyperplastic lesions that spontaneously resolve after the development of a host immune response. In contrast to the other domestic species, cats rarely develop skin warts and only two PV-induced cutaneous papillomas have been reported in domestic cats. One described a focal area of epithelial hyperplasia on the eyelid of a cat. A PV etiology in this case was suggested by the presence of PV L1 immunostaining within the lesion [70]. The second case was a papilloma that developed on the nasal planum of a cat [71]. Sequences from human PV type 9 were the only PV DNA sequences amplified from the lesion. While this could suggest cross-species infection, PVs are usually highly species specific. Additionally, as people are ubiquitously asymptomatically infected by PVs, it is hard to exclude sample contamination. As the methods used to detect PV DNA from this papilloma do not detect FcaPV1, it remains possible that this PV type could have caused the papilloma, but remained undetected in the lesion. 

### 4.2. Oral Papillomas

Oral warts have been rarely reported in cats [32,72]. Whether this is because cats only rarely develop oral warts or because many oral warts are undetected in cats is unknown. Feline oral papillomas appear to be restricted to the ventral surface of the tongue and clusters of exophytic lesions develop at this location. In a report describing two cats with oral papillomas, the lesions were incidental findings in both cases. Therefore, feline oral papillomas are likely to spontaneously resolve and unlikely to cause significant disease in cats. While there are sporadic reports of canine oral papillomas chronically persisting and eventually progressing to oral SCCs [73], there is no evidence that feline oral papillomas predispose to oral SCCs [33]. In contrast to canine oral papillomas which usually develop in dogs less than 1 years old, the feline oral papillomas were reported to develop in middle-aged cats [32]. As oral papillomas in dogs are thought to develop at the time of first infection by CPV1 [5], this suggests that cats may become first infected by FcaPV1 later in life, potentially as cats are less frequently exposed to this virus. 

Histologically, the papillomas appear as foci of markedly hyperplastic folded epithelium. Papillomavirus-induced cell changes are prominent within the papillomas [32]. Initially, immunohistochemistry was used to confirm the presence of PV L1 protein within two papillomas [72] while marked intense p16 immunostaining was subsequently detected in an additional two cases [32]. To date, the associated PV type has only been investigated in two feline oral papillomas. As both of these lesions contained FcaPV1, this PV type is considered the most likely cause of feline oral papillomas (Table 1).

Papillomaviruses in the *lambdapapillomavirus* genus cause canine oral papillomas and oral papillomas in the exotic feline species including Asian lions (*Panthera leo persica*), Snow Leopards, bobcat (*Lynx rufus*), and Florida panthers (*Puma concolor*) [3]. As FcaPV1 is the only feline PV type classified within the *lambdapapillomavirus* genus, this supports the likely role of this PV in the development of feline oral papillomas.

### 4.3. Feline Viral Plaques and Bowenoid In Situ Carcinomas 

Feline viral plaques and Bowenoid in situ carcinomas (BISCs) were initially described as separate skin lesions. However, as they are both caused by PV infection and share many histological features, they are best considered to be different severities of the same disease. For simplicity, these will be all be referred to as BISCs within the following text. 

Bowenoid in situ carcinomas are uncommon skin lesions that typically develop in middle aged to older cats. Multiple pigmented or non-pigmented non-painful, non-pruritic, slightly-raised lesions up to 2 cm in diameter most commonly develop on the face, head, and neck [74]. Lesions can develop within haired or non-haired skin suggesting that exposure to sunlight is not the primary cause of these lesions. The expected behavior of BISCs is currently poorly understood. There are reports of smaller BISCs spontaneous regressing. However, some BISCs persist and progress to an invasive SCC. Initial reports suggested that immunosuppression may be important for BISC development [75]. However, most cats that have subsequently been reported with BISC have not had any identifiable immunosuppressive disease. Sphinx and Devon Rex cats are predisposed to BISCs and these cats develop these lesions at an earlier age. Additionally, when Sphinx or Devon Rex cats develop BISCs, they more rapidly progress to invasive and even metastatic SCC [40,41].

Histological examination of BISCs reveals mild to moderate epidermal hyperplasia. Much of the hyperplasia is within the deeper layers of the epidermis and there is no marked expansion of the stratum spinosum as is typically visible in a viral papilloma. Papillomavirus-induced cell changes and L1 PV immunostaining are typically present in smaller lesions, although these tend to be lost as lesions become more advanced [74,76]. As lesions progress, cells within the epidermis become disorganized and progress to intraepithelial neoplasia. The role of the PVs in the increasing cell dysplasia and subsequent neoplastic transformation is currently unknown. However, BISCs consistently contain p16 immunostaining suggesting that PV infection is able to influence cell regulation by interfering with normal pRb function [44]. 

The majority of BISCs have been associated with FcaPV2 infection with studies detecting FcaPV2 in 5 of 21 [77] and 11 of 18 [78] BISCs using consensus PCR primers and 20 of 20 [79], 14 of 14 [44], 14 of 14 [50], and 15 of 18 [29] BISCs using primers specifically designed to amplify FcaPV2. Additionally, in situ hybridization has been used to localize FcaPV2 within the proliferating cells within the BISCs [28,29]. 

While most evidence currently suggests FcaPV2 is the predominant cause of BISCs, FcaPV3, FcaPV4, and FcaPV5 have also been associated with the development of these lesions with some evidence to suggest regional differences may exist in the predominant PV type [29,53,60]. As FcaPV2 is within a different genus to the other FcaPV types, differences in the histological features and in the behavior of the lesions would be predicted. However, the differences described in BISCs have generally been subtle with differences in the PV-induced changes caused by FcaPV2 and those caused by FcaPV3 reported [53]. Additionally, there is some evidence that FcaPV3 may cause proliferation of cells deeper within the hair follicle while infection by FcaPV5 may cause proliferation of follicular structures as well as cells within sebaceous glands [60]. While additional cases need to be followed, there is some suggestion that BISCs caused by FcaPV3 may have a less aggressive biological behavior than those caused by FcaPV2 [80]. It is currently unknown whether the different FcaPV types cause hyperplasia and neoplasia by altering the same pathways within the cells. However, intense p16 immunostaining has been reported in lesions containing FcaPV2, 3, 4, 5, and 6 suggesting all five PV types can interfere with normal pRb function [61]. 

Bowenoid in situ carcinomas are, by definition, superficial lesions that are confined to the epidermis. The lack of invasion suggests that surgical excision should be curative as long as it is possible to remove all of the affected epidermis. While this may be possible in cats with small numbers of lesions, surgical excision may not be possible in cats with larger multiple lesions over the body. For these, treatments aimed at treating just the superficial layers of skin may be most appropriate. In other species, treatment using carbon dioxide laser is considered a good therapy for superficial PV-induced lesions [81]. Likewise, cryotherapy would be expected to be a good treatment for a lesion confined to the epidermis and cryotherapy is considered the most appropriate first treatment for anogenital warts in people [82]. Imiquimod cream stimulates inflammation and therefore is useful for treating superficial skin lesions [83]. This treatment was used with some success to treat BISCs in a series of cats, although some cats were reported to develop side-effects due to the treatment [76]. While this treatment was developed to treat genital warts in people, imiquimod does not have any specific activity against PVs and other treatments that preferentially destroy infected epidermis may have similar efficacy. As BISCs appear to develop due to a poorly-characterized inability of the host to control PV replication, careful evaluation for, and resolution of, any immunosuppressive conditions may be appropriate. Additionally, cats that have been treated for BISCs are probably predisposed to developing additional BISCs in the future. 

### 4.4. Feline Cutaneous Squamous Cell Carcinomas (SCCs) 

Cutaneous SCCs are common in cats. These neoplasms are typically highly invasive and cause significant morbidity and mortality in cats [84]. As in humans, the majority of cutaneous SCCs are thought to be caused predominantly by exposure to sunlight. These SCCs develop in non-haired, non-pigmented areas of the body such as the pinna, nasal planum, and eyelids. In addition to the SCCs that develop in sun-exposed areas, a smaller number SCCs develop in areas that are protected from the sun by hair or pigment. Squamous cell carcinomas typically develop due to progression from a precursor lesion. For the SCCs that develop in sun-exposed skin, progression from actinic keratosis (a sun-induced intraepithelial neoplastic lesion) is thought to be most common [84]. It is tempting to speculate that a significant proportion of the SCCs that develop in UV-protected skin develop as a progression from a BISC, although such progression has only been documented in a small number of cases [40,41]. 

The key histological feature of a SCC is the presence of invasion of the basement membrane of the epidermis by neoplastic cells [84]. The cells within a SCC are typically markedly dysplastic with loss of the normal architecture of the epidermis. Possibly due to the markedly altered structure of the epidermis, evidence of viral replication such as virally-induced cell changes or PV L1 immunostaining are very rarely visible in feline cutaneous SCCs. 

DNA sequences from FcaPV2 were first detected in feline cutaneous SCCs in 2006 [77]. In 2008, it was shown that FcaPV2 DNA was more frequently detectible in cutaneous SCCs than in normal skin from cats [79]. Subsequently, different groups from around the world have detected PV DNA in feline cutaneous SCCs [30,45,54,85]. Additional evidence of a role of PVs in the development of feline cutaneous SCCs includes the detection of a higher viral load in BISCs and a subset of SCCs than in normal skin [26]. Furthermore, FcaPV2 gene expression has been detected [26,45] and localized with feline cutaneous SCCs [30]. Adding to the evidence that PVs contribute to cancer development, intense p16 immunostaining has been reported in SCCs that contain PV DNA while no increase in p16 immunostaining is visible in the SCCs that do not have evidence of PV infection [25,43]. In a study of nasal planum SCCs, cats with p16-positive SCCs survived longer than cats with p16-negative SCCs [43]. The different biological behavior of the two groups of SCCs suggests that these cancers developed due to different cell pathways adding evidence that SCCs can be either caused by sun-exposure or by PV infection in cats [43]. Overall, current evidence suggests that PVs influence the development of approximately 30% of SCCs from UV-exposed skin and 75% of SCCs from UV-protected skin [25]. However, the relative influence of the PV infection compared to other potential co-factors remains uncertain. 

The most common PV type that is detected in PV-associated cutaneous SCCs in cats is FcaPV2. However, a smaller proportion of SCCs have been found to contain FacPV3, FcaPV4, or FcaPV6 DNA sequences [25,29,54,57,61]. As with BISCs, there is some evidence of regional variation with FcaPV3 detected most frequently in SCCs from cats in Japan [86]. Whether or not the subsequent behavior of the SCC is influenced by the type of PV that caused the lesion is currently unknown. 

Human papillomavirus types have been detected within BISCs and SCCs in some studies [78,85]. This suggests cross-species PV infection could influence the development of these cancers. However, considering the strict species specificity of most PV types and the ubiquitous PV infection of the humans collecting or processing samples, it is extremely difficult to exclude sample contamination. Currently there is no convincing evidence that human PVs can influence feline SCC development. 

While PV-associated SCCs appear to have a more favorable prognosis, current treatments used for cutaneous SCCs in cats are generally the same regardless of the cause of the SCC. The different surgical and non-surgical techniques used for these cancers is outside the scope of this review. 

### 4.5. Feline Oral Squamous Cell Carcinomas (OSCCs)

Feline OSCCs are aggressive invasive neoplasms that are almost invariably fatal [87]. Currently there are few treatment options and cats with these neoplasms have an average survival time of approximately 5 weeks [88]. While no cause of feline oral SCCs has been established [87], a proportion of human OSCCs are well established to be caused by PV infection [89]. Considering the role of PVs in human OSCCs and in feline cutaneous SCCs, multiple studies have investigated the possibility that PVs may also cause feline OSCCs. 

In the first large investigation of feline OSCCs, PV sequences were detected in 1 of 20 cancers, but within none of 20 non-neoplastic oral samples. The PV sequence amplified from the feline OSCC was from a human PV type [90]. Papillomaviral DNA sequences were amplified from 2 of 32 feline OSCCs in a subsequent study [85]. One of the amplicons could not be sequenced while the other was from a human PV type [85]. As discussed with cutaneous SCCs, it is hard to exclude sample contamination when a human PV type is detected in feline tissue. 

No PV DNA was detected in a series of 30 feline OSCCs [7] from cats in New Zealand and in another study of 7 feline OSCCs from cats in Japan [54]. In a different study of cats from New Zealand, sequences from FcaPV1 were detected in 1 of 36 OSCCs and 1 of 16 inflammatory gingival lesions [33] with no other PV types identified in this study. A recent study used next generation sequencing to identify all mammalian and viral DNA within a series of 20 feline OSCCs and 9 samples of normal feline gingiva from cats in North America. Although the samples were found to contain a variety of virus types, only one OSCC was found to contain a PV sequence, which was from FcaPV3 [55]. In contrast, in a study of Italian cats, FcaPV2 was detected in 10 of 32 (31%) feline OSCCs and 4 of 11 (36%) samples of ulcerative gingivitis [91]. In addition, gene expression was detected in many of the positive samples suggesting that viral replication was present in the oral samples [91]. Viral loads were compared between the OSCCs and ulcerative gingivitis samples, with no significant differences detected. In situ hybridization to confirm the location of the FcaPV2 DNA in the samples was inconclusive in the study [91]. Sequences from FcaPV2 DNA were also detected in 11 of 19 (58%) OSCCs in a recent study of cats from Taiwan. No non-neoplastic oral samples were included in this study [86].

Multiple studies have reported variable p16 immunostaining within feline oral SCCs. However, this variable immunostaining has not been associated with the presence of PV DNA in feline OSCCs [7,46,54,91]. This is consistent with the variable p16 immunostaining being due to spontaneous mutations within the neoplasms rather than due to a PV etiology. 

Overall, significant differences in the detection of PV DNA within feline oral SCCs have been reported by different research groups. Whether this represents a geographical difference in the tissue tropism of FcaPV2 or reflects differences in the techniques used is currently unknown. However, as similar rates of detection of PV DNA have been reported in OSCCs and non-neoplastic samples [91] and p16 immunostaining has not been associated with the presence of PV DNA [7,46,91], there is currently no evidence to support a causative association between PVs and feline OSCCs.

### 4.6. Feline Basal Cell Carcinomas (BCCs)

Basal cell carcinomas are much less common than SCCs in cats. Unlike SCCs, these neoplasms do not have a connection to the overlying epidermis and keratinization is generally absent. The neoplastic cells are small and darkly basophilic and so resemble cells within the basilar layers of the epidermis [84]. 

The possibility that BCCs are associated with PV infection in cats was first reported after it was observed that these lesions are often associated with adjacent BISC lesions and the observation of PV cytopathic changes in cells within some BCCs [92]. Subsequently, a cat with multiple BCCs that contained FcaPV3 DNA was reported [53]. Additionally, short sequences from a novel PV type were amplified from a feline cutaneous BCC [93]. The PV type from which these sequences were derived has not been fully sequenced suggesting that at least one additional PV types will be detected from cats in the future. 

### 4.7. Feline Sarcoids

Feline sarcoid are different from the other PV-associated pre-neoplastic and neoplastic lesions described in this review because the proliferating cells are mesenchymal rather than epithelial. Additionally, in contrast to the previously discussed lesions, feline sarcoids are thought to be caused by cross-species infection by a bovine PV [62,68]. Infection by this BPV appears to require close contact between cattle and cats. For this reason feline sarcoids are rare and most often seen in cats that live on dairy farms [94].

Sarcoids most often develop as firm, exophytic masses on the nasal philtrum upper lips, digits, or mouth of young to middle-aged cats from rural areas [94,95]. Histologically, sarcoids appear as a proliferation of haphazardly arranged collagen bundles covered by hyperplastic epidermis. Feline sarcoids do not appear to permit viral replication and they do not contain PV-induced cell changes or PV L1 immunostaining. Evidence supporting BPV14 as the cause of feline sarcoids includes the consistent detection of this PV in feline sarcoids, and the consistent absence of BPV14 in non-sarcoid samples from cats [65]. Additionally, PV DNA has been localized in the neoplastic mesenchymal cells using in situ hybridization [96]. Therefore, current evidence suggests that BPV14 causes feline sarcoids and the detection of this PV within a feline mesenchymal tumor confirms a diagnosis of feline sarcoid. Whether infection of a cat by this PV type always results in sarcoid development or whether co-factors are needed for sarcoid formation is currently unknown. 

Complete surgical excision is recommended to treat feline sarcoids. In a recent study, approximately 10% of sarcoids recurred after surgical excision that was considered to be complete by histology. In contrast, 67% of sarcoids recurred after being histologically classified as incompletely excised with many of these cats eventually being euthanatized due to local tumor effects [97]. Currently, no medical therapies have been reported to be effective in the treatment of feline sarcoids. 

### 4.8. Other Cancers of Cats

The ability of high-risk human PVs to cause oral and anogenital SCCs is well established. However, PVs have also been suggested to influence the development of lung, breast, and bladder cancers in people [98,99,100]. In a study of cats, no PV DNA was amplified from a series that contained nine lung, nine mammary gland, and five bladder cancers [101]. This does not support a PV etiology of these cancers, although as only small numbers of samples were included, a role of PVs cannot be excluded. 

## 5. Prevention of Papillomaviral Disease in Cats

A proportion of feline cancers appear to be caused by PVs. Therefore, these cancers could be prevented by preventing infection by the causative PV. In human medicine, PV vaccines are used to prevent PV-induced papillomas (warts) as well as PV-induced genital and oral cancers [102]. While vaccines have been highly successful in humans, the use of vaccines faces some significant challenges in cats. Firstly, to be effective a vaccine has to be given prior to first infection. This is easy for the human PV vaccines as the causative PV types are spread by sex, providing an ample window of time in which to vaccinate. In contrast, cats have been shown to be infected by FcaPV2 within the first few days of life [21] making it difficult to vaccinate cats prior to first infection. Recently, a virus-like-particle vaccine against FcaPV2 was developed and used in adult cats [37]. Due to the ubiquitous nature of FcaPV2 infection all cats were already infected by FcaPV2 at the time of vaccination. While vaccination resulted in a 7-fold increase in antibody titres in cats, the increased antibodies did not reduce viral load. If the viral load influences which cats will develop lesions due to FcaPV2, these results suggest that vaccinating a cat that is already infected by FcaPV2 will not influence disease development [37]. In an experimental setting it may be possible to vaccinate kittens prior to being first infected. However, these vaccines do not appear to be a practical way to prevent PV infection in more general veterinary medicine. The lack of reduction in viral loads due to vaccination also suggests that a FcaPV2 vaccine would not be useful as a treatment [37]. In people, initial evidence from a number of studies suggested that vaccines could be useful in treating PV-induced warts [103]. However, in a recent study of approximately 500 patients, the use of HPV vaccines was not found to have any statistically significant benefit in the treatment of anogenital warts [104].

While the majority of PV-induced diseases appear to be caused by FcaPV2 in cats, vaccines could be developed against other PV types that infect cats. Infection with BPV14 appears to be rare in cats, suggesting it would be possible to vaccinate against this PV type prior to first infection [65]. As feline sarcoids can result in death of the cat [68], protecting against infection would be valuable. However, feline sarcoids are very rare feline tumors that are limited to cats that have close contact to cattle. Considering the small minority of cats that are susceptible to feline sarcoids and the overall rarity of these tumors, it appears unlikely there would be a commercial market for a vaccine against BPV14. 

It is currently unknown when cats are infected by FcaPV1, although evidence from dogs suggests that the oral papillomas probably develop at the time of first infection. If most cats are not infected until adulthood, this would suggest a potential window in which to vaccinate. However, as with feline sarcoids, cats appear to develop these lesions only rarely [32]. In addition, in the small numbers of cats in which papillomas have been detected, these were detected as incidental findings and there is no evidence that oral papillomas cause significant disease in cats [32]. 

It is also currently unknown when cats are infected with the *Taupapillomaviruses* FcaPV3, 4, 5, and 6. It is possible that infection is later in life, allowing an opportunity for vaccination. However, each of these PV types are currently thought to be rare causes of disease in cats. Furthermore, a vaccine would need to be given for each type as the currently used PV vaccines do not provide cross-protection against different PV types [102]. The need to include multiple virus types would increase the cost of any vaccine and, considering the rarity of disease caused by these PV types, a vaccine is unlikely to be commercially viable. 

## 6. Conclusions

While domestic cats have been comparatively recently recognized to develop clinical disease due to PVs, knowledge of the range of PV types that infect cats and the different lesions that are caused by infection has expanded rapidly in recent years. It appears likely that additional FcaPV types will be identified in the future and it is possible that a role of PVs in other diseases in cats may be identified. Currently, preventing PV infections in cats appears to be difficult. As all cats are infected by FcaPV2 and potentially the other FcaPV types, it may be better to investigate ways of preventing a normal asymptomatic infection from progressing to a hyperplastic or even neoplastic disease. To do this, the underlying changes to the host that allow proliferation of PVs and subsequent disease development need to be better understood. Thus, the aim of future research may be to better understand the critical interactions between the feline host and PVs. 

## Figures and Tables

**Figure 1 viruses-13-01664-f001:**
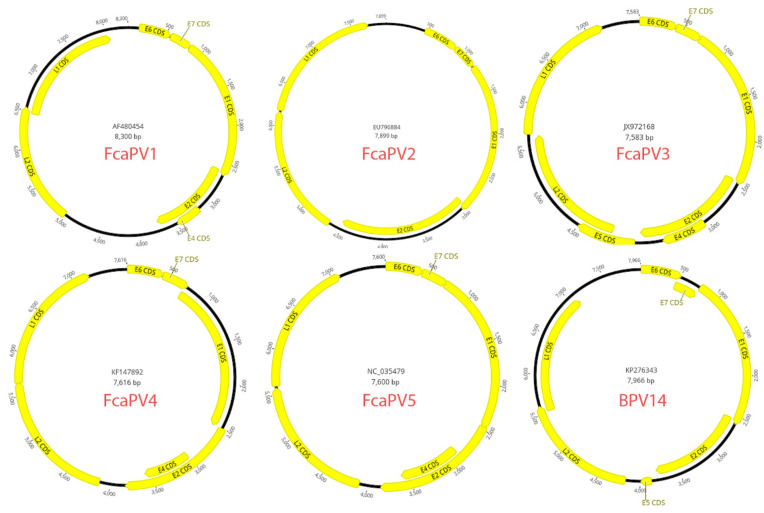
Schematic representation of the genetic arrangement of six of the seven papillomavirus types that are known to infect cats. Figures created using Geneious version 2021.2 created by Biomatters. Available from https://www.geneious.com (accessed on 15 August 2021).

**Figure 2 viruses-13-01664-f002:**
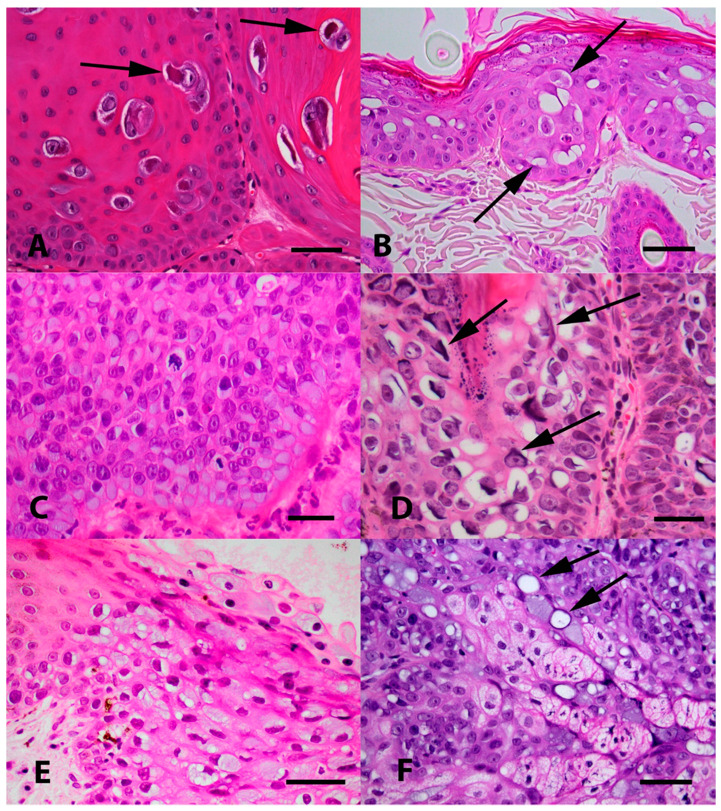
(**A**) Photomicrograph of an oral papilloma caused by Felis catus papillomavirus type 1 (FcaPV1). Infection of cells by this papillomavirus (PV) tends to result in the presence of a relatively small proportion of cells developing expanded cytoplasm and nuclei. Within the nuclei, the chromatin becomes compressed peripherally with the presence of a prominent nucleoli. However, the most characteristic feature of FcaPV1 is the development of large prominent eosinophilic cytoplasmic bodies within the expanded cytoplasm (arrows). HE Bar = 30 µm. (**B**) Photomicrograph of an early Bowenoid in situ carcinoma caused by FcaPV2. Infected cells have expanded cytoplasm that is wispy and blue/grey or clear (arrows). Nuclei are enlarged, but do not have the central chromatin clearing as is visible within cells infected by FcaPV1. Cells within deeper layers of the epidermis have dark shrunken nuclei that are surrounded by a clear halo (koilocytosis). HE Bar = 22 µm. (**C**) Photomicrograph of a more advanced Bowenoid in situ carcinoma caused by FcaPV2. Many of the cells have expanded cytoplasm that has an amorphous texture and is blue/grey. There is loss of normal epidermal maturation consistent with the diagnosis of intraepithelial carcinoma. HE Bar = 23 µm. (**D**) Photomicrograph of a Bowenoid in situ carcinoma caused by FcaPV3. Cells in this lesion are expanded, but the cytoplasm is generally clear. Nuclei are enlarged but lack the central clearing. A prominent feature seen in cells infected by FcaPV3 is the presence of large elongated amphophilic to basophilic cytoplasmic bodies (arrows). These are often present adjacent to the nuclear membrane of the cell. HE Bar = 22 µm. (**E**) Photomicrograph of an area of oral epithelial hyperplasia that contained FcaPV4 DNA sequences. Although additional lesions containing this PV type are required to make definitive conclusions, infection has resulted in increased blue-grey slightly granular cytoplasm. Nuclei do not appear to be markedly enlarged and are typically displaced to the periphery of the cell. HE Bar = 25 µm. (**F**) Photomicrograph of a Bowenoid in situ carcinoma associated with FcaPV5. The cells that were infected in this lesion were close to the base of the hair follicle and within sebaceous glands. While additional cases are required, the presence of sebaceous glands that have blue-grey expanded cytoplasm should be considered suggestive of infection by this PV type. Another feature visible is the presence of cells that have a single large cytoplasmic vacuole that has compressed the remainder of the cytoplasm to the periphery (arrows). HE Bar = 35 µm.

**Table 1 viruses-13-01664-t001:** Summary of lesions associated with each papillomavirus type known to infect cats. SCC is squamous cell carcinoma.

Papillomavirus Type	Classification	Associated Lesion
FcaPV1	*LambdaPV*	Oral papillomas
FcaPV2	*DyothetaPV*	Bowenoid in situ carcinoma, cutaneous SCC
FcaPV3	*TauPV*	Bowenoid in situ carcinoma, cutaneous SCC, basal cell carcinoma
FcaPV4	*TauPV*	Bowenoid in situ carcinoma, cutaneous SCC
FcaPV5	*TauPV**	Bowenoid in situ carcinoma
FcaPV6	*TauPV**	Cutaneous SCC
BPV14	*DeltaPV*	Feline sarcoid

## Data Availability

All data contained within the review is accessable using publically-searchable article databases.

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
