# Peer review of "Papillomaviruses in Domestic Cats"

_viruses, 2021, doi:10.3390/v13081664_

Round 1

Reviewer 1 Report

Review: "Papillomaviruses in domestic cats"
by John S Munday and Neroli Thomson.

The authors summarise the current state of research concerning feline papillomaviruses. The manuscript describe four aspects of papillomaviruses infecting cats: First, the methods to assess an infection in cats, second, the different papillomaviruses infecting cats, third, the typical lesions observed, fourth, a discussion of prevention against papillomaviruses in cats. The paper is interesting and clearly written. I strongly suggest to accept this manuscript for publication in Viruses. There are only a few details which could be addressed:

- scale bars should be included on the micrographs of Figure 1. The letters on the micrographs are not well visible. A brighter font color or outlined letters might solve this problem. 

- BPV13 might be mentioned as delta-PV causing sarcoids in horses on line 308. 

- it would be helpful to summarise the papillomaviruses described in the text as table including taxonomic classification and related disease. 

Author Response

Review: "Papillomaviruses in domestic cats"

by John S Munday and Neroli Thomson.

The authors summarise the current state of research concerning feline papillomaviruses. The manuscript describe four aspects of papillomaviruses infecting cats: First, the methods to assess an infection in cats, second, the different papillomaviruses infecting cats, third, the typical lesions observed, fourth, a discussion of prevention against papillomaviruses in cats. The paper is interesting and clearly written. I strongly suggest to accept this manuscript for publication in Viruses. There are only a few details which could be addressed:

Thank you very much for the positive comments.

- scale bars should be included on the micrographs of Figure 1. The letters on the micrographs are not well visible. A brighter font color or outlined letters might solve this problem.

Scale bars have been added as requested and letters in the figures have been made bold to make them easier to see.

- BPV13 might be mentioned as delta-PV causing sarcoids in horses on line 308.

This has been added within the text (line 322).

- it would be helpful to summarise the papillomaviruses described in the text as table including taxonomic classification and related disease.

The authors agree this is a good idea and a table has been added as suggested (table 1).

Reviewer 2 Report

The manuscript entitled „ Papillomaviruses in domestic cats “is a review submitted by the scientist who study animal papillomaviruses quite intensely. They have written review in 2017, “Papillomaviruses in dogs and cats”. Some of the information in this present review is overlapping with the information in the previous review what is unavoidable but information about the newly discovered cat PVs and their characteristics are included.  

The manuscript to my opinion would benefit from adding a figure with the structure of the feline papillomavirus genomes and more detailed description of genomes will be useful for readers.  For clarity also, Table summarizing basic information about the feline PV types can be included. Additionally, also the information about treatment of PV caused lesions in cats can be mentioned in this review.                                                                                                                                                                                                           

Specific comments:

FcaPV3 and 4 belongs to Taupapillomaviruses 3, not 2 (Papillomaviridae - Papillomaviridae - dsDNA Viruses - ICTV (ictvonline.org).

On several places I suggest to add quoting.

On page 2, line 84, citation is missing, as well as citation to a sentence on page 2, line 85-86.

Another citation missing is on page 3, line 139 (consensus primers).

In the description of discrete cat PVs, no quoting for the description of the histological features is present. If this is personal observation of the authors it should be mentioned.

On page 8, line 287 quoting is missing.

On page 8, line 326-327 the statement “PCR was used to amplify human PV type 9 DNA sequences from this papilloma suggesting cross-species infection„ seems too strong for the available data.

Similarly, the statement „The potential involvement of FcaPV1 in the development of the nasal planum papilloma is supported by the observation that the methods used to detect PV DNA in the lesion do not detect FcaPV1„ is not to my opinion a correct justification for the link of FcaPV1 with the specific lesions.

Quoting on page 11, line 464 (83) should be replaced by a more appropriate one.

I suggest to delete on page 12 chapter “other cancers of cats”, because the justification by the example of other cancers in humans linked to PV infection is not appropriate. These types of cancers were never proved to be etiologically linked to HPV infection.

On page 13, line 560, “This is consistent with a recent study of human patients that also found vaccination after the development of anogenital warts did not significantly influence wart resolution [96] “ is not appropriate quoting. In humans, even though not completely understood, it has been admittedly show that prophylactic vaccination decreases the frequency of recurrences.

In conclusion the manuscript can be recommended for publication after revision.   

Author Response

The manuscript to my opinion would benefit from adding a figure with the structure of the feline papillomavirus genomes and more detailed description of genomes will be useful for readers.  For clarity also, Table summarizing basic information about the feline PV types can be included. Additionally, also the information about treatment of PV caused lesions in cats can be mentioned in this review.                                                                                                                                                                                                          

As suggested figures demonstrating the genetic structures of examples from each of the PV genera have been included. Additionally, a table has been added to summarize the basic information about each PV type and which lesions each type has been associated with. Where appropriate, information regarding treatment has been added to the manuscript (lines 424-442).  

Specific comments:

FcaPV3 and 4 belongs to Taupapillomaviruses 3, not 2 (Papillomaviridae - Papillomaviridae - dsDNA Viruses - ICTV (ictvonline.org).

Thank you very much for highlighting this error – this has been changed throughout the text (lines 239, 270, 307).

On several places I suggest to add quoting.

On page 2, line 84, citation is missing, as well as citation to a sentence on page 2, line 85-86.

References for these statements have been added (line 83).

Another citation missing is on page 3, line 139 (consensus primers).

References for the commonly used consensus primers have been added (line 142).

In the description of discrete cat PVs, no quoting for the description of the histological features is present. If this is personal observation of the authors it should be mentioned.

References have been added as suggested (lines 168, 229, 245. 263, 276).

On page 8, line 287 quoting is missing.

This reference has been added (line 417).

On page 8, line 326-327 the statement “PCR was used to amplify human PV type 9 DNA sequences from this papilloma suggesting cross-species infection„ seems too strong for the available data.

Similarly, the statement „The potential involvement of FcaPV1 in the development of the nasal planum papilloma is supported by the observation that the methods used to detect PV DNA in the lesion do not detect FcaPV1„ is not to my opinion a correct justification for the link of FcaPV1 with the specific lesions.

The information about this papilloma has been rewritten according to the reviewer’s suggestions (line341-346).

Quoting on page 11, line 464 (83) should be replaced by a more appropriate one.

This reference has been replaced as suggested (line 502).

I suggest to delete on page 12 chapter “other cancers of cats”, because the justification by the example of other cancers in humans linked to PV infection is not appropriate. These types of cancers were never proved to be etiologically linked to HPV infection.

The authors would prefer to leave this section in as there remains considerable interest in humans about the role of PVs in other cancers. The authors agree that the evidence for a PV cause is not strong and the wording of this section has been modified to reflect the uncertainty (line 585). A single study also found no evidence of PV involvement in feline neoplasms which is an important result in that it will stop others from looking for PVs in other cancers in cats.

On page 13, line 560, “This is consistent with a recent study of human patients that also found vaccination after the development of anogenital warts did not significantly influence wart resolution [96] “ is not appropriate quoting. In humans, even though not completely understood, it has been admittedly show that prophylactic vaccination decreases the frequency of recurrences.

The authors agree there remains uncertainty regarding whether vaccines can influence wart resolution. This has been better communicated by adding a reference that reviewed earlier studies the often found a beneficial effect of vaccines. Additionally, more background information is included regarding the recent large study that did not report any significant effect of vaccines (lines 608-612).